# Glucose and Cell Context-Dependent Impact of BMI-1 Inhibitor PTC-209 on AKT Pathway in Endometrial Cancer Cells

**DOI:** 10.3390/cancers14235947

**Published:** 2022-12-01

**Authors:** Agnieszka Zaczek, Aleksandra Szustka, Anna Krześlak

**Affiliations:** 1Department of Cytobiochemistry, Faculty of Biology and Environmental Protection, University of Lodz, 141/143 Pomorska St., 90-236 Lodz, Poland; 2Department of Molecular Biophysics, Faculty of Biology and Environmental Protection, University of Lodz, 141/143 Pomorska St., 90-236 Lodz, Poland

**Keywords:** BMI-1, AKT pathway, glucose, insulin, EMT, endometrial cancer cells

## Abstract

**Simple Summary:**

The insulin/IGF-1/AKT pathway is a key event linking metabolic syndrome with endometrial cancer. It has been found that BMI-1 may also influence the AKT kinase pathway and that increased activity is associated with cancer onset and progression. The aim of our study was to determine the role of BMI-1 in glucose and molecular context-dependent regulation of PHLPP expression and AKT kinase activity in endometrial cancer cells. The studies were conducted using endometrial cancer cells (HEC-1A, Ishikawa). Glucose and insulin impact on the BMI-1 dependent regulation of PHLPP-analyzed cells with the downregulation of BMI-1 expression in hypo-, normo-, and hyperglycemia conditions and stimulation by insulin. Understanding the relationship between hyperglycemia, BMI-1, PTEN, and PHLPP functions, may contribute in the future to the development of therapeutic approaches that will be better adapted to the molecular context and the specificity of endometrial cancers and more effective in cases of metabolic diseases co-existence.

**Abstract:**

Purpose: In our study, the glucose and cell context-dependent impact of the BMI-1 inhibitor PTC-209 on the AKT pathway in endometrial cancer cells was determined. Methods: The expression of BMI-1 was inhibited by PTC-209 in endometrial cancer cells HEC-1A and Ishikawa stimulated with insulin and grown in different glucose concentrations. The migration, invasion, viability, and proliferative potential after PTC-209 treatment was assessed using wound-healing, Transwell assay, Matrigel-coated inserts, and MTT tests. Chromatin immunoprecipitation was used to determine the localization of BMI-1 protein at promoter sites of the genes tested. Results: BMI-1 inhibition caused an increase in *PHLPP1*/2 expression and a decrease in phospho-AKT level in both cell lines. The glucose concentration and insulin stimulation differentially impact the AKT pathway through BMI-1 in cells differing in PTEN statuses. The expression of BMI-1 is dependent on the glucose concentration and insulin stimulation mostly in PTEN positive HEC-1A cells. In high glucose concentrations, BMI-1 affects AKT activity through PHLPPs and in hypoglycemia mostly through PTEN. BMI-1 inhibition impacts on genes involved in *SNAIL*, *SLUG*, and *CDH1* and reduces endometrial cancer cells’ migratory and invasive potential. Conclusions: Our results indicate that the relationship between BMI-1 and phosphatases involved in AKT regulation depends on the glucose concentration and insulin stimulation.

## 1. Introduction

Epidemiologic data suggest that obesity, hyperglycemia, and hyperinsulinemia are serious risk factors for endometrial cancer [1,2]. Insulin resistance is a characteristic feature of metabolic syndrome, which occurrence is associated with more advanced cancers and worse prognosis in the case of endometrial cancer patients [2]. An insulin pathway is a key event linking metabolic syndrome with cancer. Insulin binds to tyrosine kinase receptors on the surface of the classic insulin-responsive cells, mainly the hepatocytes, adipocytes, and muscle cells, which express high levels of the insulin receptor. However, the IR (Insulin Receptor) is also expressed in other tissues including the endometrium and breast tissues [3,4]. Insulin can also stimulate IGF-1R (Insulin-like Growth Factor 1 Receptor). The IGF-1R shows a high degree of homology with the insulin receptor and hybrid IGF-1R/insulin receptor species have been reported in cancers [5]. Insulin through stimulation of the cell surface receptor tyrosine kinase (RTK) can activate AKT kinase. The AKT pathway plays important role in cell metabolism, growth, proliferation, and survival. RTK activates the PI3K kinase, which converts PIP2 (phosphatidylinositol-3,4-triphosphate) to PIP3 (phosphatidylinositol-3,4,5-triphosphate). PIP3 contributes to the recruitment of AKT kinase to the cell membrane where it is phosphorylated [6]. Several phosphatases play important role in AKT dephosphorylation. PTEN, INPP4B (inositol polyphosphate 4-phosphatase type II), and INPP5D (Inositol Polyphosphate-5-Phosphatase D) are phosphatases that indirectly affect AKT phosphorylation by dephosphorylation of the second messenger PIP3 to PIP2 [7,8]. PP2A (Protein phosphatase 2A) and PHLPP1/2 are involved in direct AKT isoform dephosphorylation on Thr308 and Ser473, respectively [9,10]. It was shown that the activity of the AKT pathway might also be affected by B lymphoma Mo-MLV insertion region 1 homolog (BMI-1), which is a component of the PRC1 complex that regulates gene transcription via ubiquitylation of histone H2A (Lys119) [11].

BMI-1 may affect AKT activity indirectly by PTEN regulation [12,13]. Moreover, the results of our previous studies on endometrial cancer suggested that BMI-1 may be involved in the regulation of PHLPP [14]. PHLPP belongs to a novel family of Ser/Thr protein phosphatases that consists of PHLPP1 and PHLPP2 isoforms. Increasing evidence indicates that PHLPP isoforms serve as tumor suppressors. Conversely, the level of PHLPP1, but not PHLPP2, is elevated in diabetes [15]. Reduced PHLPP expression, which is a negative regulator of AKT, may be one of the main reasons for increased activity of the AKT pathway and promotion of cancer cell survival. Alterations in the expression and activity of BMI-1 are responsible for the incorrect expression of many other genes that control cell division, proliferation, apoptosis, and cell migration leading to cancer onset and progression [16]. Although BMI-1 is a known oncogene contributing to the initiation of malignant transformation and increased cell proliferation, its role in invasion and metastasis is not fully established. It has been found that both low and high BMI-1 levels might contribute to the progression of cancer and have an impact on increased cell ability for migration and invasion [12,17,18,19,20].

This study aimed to determine the role of BMI-1 inhibition by PTC-209 in the regulation of PHLPPs expression and AKT kinase phosphorylation in high and low conditions in endometrial cancer cells differing in PTEN status and the influence of BMI-1 on migratory and invasion potential of these cells. Our study suggested that depending on glucose availability and molecular context, BMI-1 can differently affect the activation of the AKT pathway by regulation of PHLPP isoform expression in endometrial cancer cells.

## 2. Material and Methods

### 2.1. Cell Culture and Treatment

The experiments were conducted on endometrial cancer cell lines HEC-1A (American Type Culture Collection, Manassas, VA, USA) and Ishikawa (European Collection of Aunthenticated Cell Cultures, Wiltshire, UK). Both of the endometrial cell lines were grown in DMEM:F12 media (Biowest, France) containing 10% (HEC-1A) or 5% (Ishikawa) (*v*/*v*) FBS in standard condition (37 °C, 5% CO_2_).

HEC-1 and Ishikawa cell lines were treated with 0.5, 1, or 5 µM PTC-209 (MedChem Express, Monmouth Junction, NJ, USA) under different conditions. The endometrial cancer cells were treated with PTC-209 in standard culture conditions or low (0.5 mM) or high (30 mM) glucose concentrations. The effect of the treatment was checked after 48 h. Additionally, the cells grown in low or high glucose after 24 h of serum starvation were stimulated with 100 nM insulin (Sigma Aldrich, St. Louis, MO, USA) and were harvested after 2, 4, 6, and 8 h.

### 2.2. RNA Extraction, cDNA Synthesis, and Real-Time Quantitative PCR

The total RNA from cells were isolated using a Tissue Total RNA GPB Mini Kit (Genoplast Biochemicals, Pruszków, Poland) following the manufacturer’s protocol and quantified spectrophotometrically. After, the RNA isolation reverse transcription reaction was performed using a High-Capacity cDNA Reverse Transcription Kit (Applied Biosystem, Waltham, MA, USA), according to the protocol included in the kit. TaqMan^®^ Gene Expression Assay (ThermoFisher Scientific, Waltham, MA USA) was used for the amplification of the cDNA. The Appendix A show the fluorogenic, FAM-labeled probes, and specific sequences of primers for *BMI1*, *PTEN*, *PP2A, PHLPP1*, *PHLPP2*, *SNAIL, SLUG, ZEB1, TWIST, CDH1,* and the internal control *HPRT1.*

The fold differences in gene expression, normalized to *HPRT1* levels were calculated using the formula 2^ΔΔCt^.

### 2.3. Chromatin Immunoprecipitation Assay

The chromatin immunoprecipitation was used to determine the localization of the BMI-1 protein at promoter sites of the PTEN, PHLPP1, and PHLPP2 genes in endometrial cancer cells HEC-1A and Ishikawa cultured for 48 h with or without PTC-209. Following treatment, the formaldehyde was added and the cells were incubated at room temperature for 10 min. The cross-linking reactions were terminated by adding glycine to a final concentration of 0.125 M. The cells were washed with PBS and lysed with Lysis Buffer (5 mM PIPES pH 8.0, 85 mM KCl, 0.5% NP-40, and Protease Inhibitor Cocktail) to collect nuclei. The nuclei were pelleted with Lysis Buffer High Salt (1 X PBS, 1% NP-40, 0.5% Sodium Deoxycholate, 0.1% SDS, and Protease Inhibitor Cocktail) and sonicated on ice using Vibra Cell TM model VCX-130. After, sonification immunoprecipitation of chromatin fragments was performed with specific antibodies against BMI-1. As a control of immunoprecipitation and non-specific precipitation, normal IgG was used. The protein and antibody complexes conjugated to Protein A/G Plus agarose beads. The real-time quantitative PCR was used to quantify differential chromatin enrichment.

The primer pairs that covered the *PTEN*, *PHLPP1,* and *PHLPP2* regions were as follows:

*PTEN*: 5′ CGGGCGGTGATGTGGC 3′ and 5′ GCCTCACAGCGGCTCAACTCT 3′.

*PHLPP1:* 5′AGACGGGGCCAGCGATCCTGTGAA3′ and 5′GTCGAGGATACCCAGAAGA3′.

*PHLPP2:* 5′ATGTGGTTTCATGTGTTTGTTCTCA3′ and 5′CATGGCTTTGTTTTAAAATGGAGTG 3′.

### 2.4. Western Blot and Densitometric Analysis

The RIPA buffer (50 mM Tris HCl pH 8, 150 mM NaCl, 0.5% sodium deoxycholate, 1% Nonidet P-40, 0.1% SDS, 1 mM EDTA, 1 mM PMSF) was used to lyse the endometrial cancer cell. The protein concentration in cell lysates was determined using the Lowry method. The separation of protein in cell lysates was performed using 8% SDS-PAGE. To detect proteins on immunoblots, specific primary antibodies were used: anti-BMI-1 (#6964, Cell Signaling Technology, USA), anti-PTEN (sc-7974, Santa Cruz Biotechnology, Santa Cruz, CA, USA), anti-phospho-Akt (Ser473) (#9271, Cell Signaling Technology, Beverly, MA, USA), and anti-AKT (sc-5298, Santa Cruz Biotechnology, Santa Cruz, CA, USA). Goat anti-mouse or anti-rabbit secondary antibodies conjugated with horseradish peroxidase (Cell Signaling Technology, Beverly, MA, USA) were used as secondary antibodies. For loading, the control detection of β-actin was performed (anti-β-actin antibody (sc-477778, Santa Cruz Biotechnology, Santa Cruz, CA, USA). The bands visualized by immunodetection were analyzed by densitometry using a Gel-Pro Analyzer software version 3.0 (Media Cybernetics, Inc., Bethesda, MD, USA). The IOD (integrated optical density) obtained in the Gel Pro program was used to estimate relative protein expression. The relative protein level is presented as a ratio of the IOD of the bands corresponding to the analyzed protein in each sample/IOD of β-actin in the same sample. The phospho-AKT/AKT ratio is shown as the AKT phosphorylation level.

### 2.5. MTT Assay

The proliferation of HEC-1A/Ishikawa was assessed by measuring the ability of live cells to metabolize 3-(4,5-dimethylthiazolo-2-yl)-2,5-diphenyl tetrazolium bromide (MTT) to formazan. The cells were seeded onto 96-well plates at a density of 9 × 10^3^ cells per well (HEC-1A) or 7 × 10^3^ per well (Ishikawa). On the next day, the cells were treated with 0.5, 1, or 5 µM PTC-209 in hypo- or hyperglycemia conditions. At the end of treatment, a fresh medium containing MTT was added to the cell monolayers and, after 2 h, the medium was removed and formazan crystals were dissolved in DMSO. The absorbance was read at 590 nm.

### 2.6. Migration/Invasion Assay

To assess the rate of migration or invasion of endometrial cancer cells treated with 5 µM PTC-209, the Transwell assay was performed using cell culture inserts (polyethylene terephthalate PET membranes with 8 μm pores) (Greiner Bio-One, Kremsmünster, Austria). After 24 h treatment, the cells were plated in a serum-free medium and placed in the upper chamber. The lower chamber contained a medium with serum. The cells were incubated for 24 h and after that cells were removed from the upper chamber. The migrated cells at the bottom of the insert were fixed in 4% paraformaldehyde and stained with Giemsa. The assay chambers were coated with Matrigel^®^ Matrix Basement Membrane (Corning, Corning, NY, USA) in case of invasion.

### 2.7. “Wound-Healing” Assay

The cell lines were seeded at density 4 × 10^5^ (HEC-1A) or 3 × 10^5^ into a 12-well plate in the standard conditions. After completely removing DMEM, the adherent cell monolayer was wounded by a manual scratch with a sterile 200 μL pipette tip. The phosphate Buffer Saline (PBS) was used for cellular debris removing.

The cells were incubated in standard conditions for 48 h with or without PTC-209 and then the recording of images of the scratch area was carried out at three different points using a Nikon Eclipse TE200 microscope with Zeiss CCD video camera AcioCam ERc5s at 0 h (just after scratching cells), at 24 h and 48 h. To determine the migration potential of the cells, the size of the scratch was measured using a light microscope scale. The graphs were created based on the scratch values measured. The size of the scratch at the time T0 is recorded as 100%.

### 2.8. Statistical Analyses

The statistical analyses were performed using a GraphPad Prism 5.0 program (GraphPad Software Inc., San Diego, CA, USA). To compare the differences between the treated and untreated cells, the Student paired *t*-test was used. A statistically significant value was assumed for *p*-value < 0.05.

## 3. Results

### 3.1. Inhibition of BMI-1 Expression in Endometrial Cancer Cells

The results of our previous studies showed that BMI1 silencing caused a reduction in AKT phosphorylation and increased the expression of gene coding for *PHLPP1* and *PHLPP2* phosphatases in endometrial cancer cells HEC-1A and breast cancer MDA-MD-23 cells [14]. In this study, to decrease BMI-1 protein level in HEC-1A cells and Ishikawa cells, we used the PTC-209 inhibitor, which is a potent and selective inhibitor that effectively reduces the synthesis of BMI-1 protein. The results confirmed its effectiveness because in HEC-1A and Ishikawa cells treated with PTC-209 BMI-1, the protein level was substantially reduced (Figure 1A). In both cell types treated with PTC-209, the level of AKT phosphorylation was decreased and the expressions of PHLPPs were increased (Figure 1A,B). These results are similar to results obtained previously for HEC-1A [14] and Ishikawa cells (Appendix A) treated with siRNA. In Ishikawa cells after PTC-209 treatment, the increased expression of the *PTEN* transcript was observed but there was no expression of PTEN at the protein level (Figure 1C). It has been shown earlier that Ishikawa cells do not possess functional PTEN protein because of deletions in the PTEN transcript deprive proper protein synthesis [21]. Our results also showed that there is no PTEN protein in Ishikawa cells (Figure 1C). The ChIP assay with BMI-1 specific antibodies was used to check the binding of BMI-1 to promoter regions of the *PTEN*, *PHLPP1,* and *PHLPP2* in HEC-1A and Ishikawa cells (Figure 1D,E). The results showed that, in cells treated with PTC-209, the anti-BMI-1 antibody binding level was lower compared to control cells (Figure 1D,E).

### 3.2. Different Impact of BMI-1 Inhibitor on Phosphatases Expression in Endometrial Cancer Cells in Hypoglycemia and Hyperglycemia Conditions

To find out whether the glucose concentration has an impact on BMI-1 expression and phosphatases regulation, the HEC-1A and Ishikawa cells were grown in hypo- (0.5 mM) and hyperglycemia (30 mM) conditions. In HEC-1A cells the PTEN and PHLPP1 transcript levels were lower in 30 mM glucose than in 0.5 mM glucose (Figure 2A). In Ishikawa cells, the only level of PHLPP1 transcript was significantly lower. Interestingly, the protein level of BMI-1 was higher in both cell types growing in low glucose compared to cells growing in high glucose (Figure 2B). In hyperglycemia, in both cell types, the pAKT/AKT ratio decreased after PTC-209 treatment. However, after PTC-209 treatment upon hypoglycemia conditions in HEC-1A cells, the pAKT/AKT ratio decreased and in Ishikawa cells increased (Figure 2B). The substantial differences in phosphatases expression were found depending on whether BMI-1 depletion by PTC-209 was in low or high glucose (Figure 2C,D). In hyperglycemia conditions, the downregulation of BMI1 expression caused a significant increase in *PHLPP1* and *PHLPP2* in both cell lines. However, in low glucose concentrations, there was a significant decrease in *PHLPP1* in BMI-1 depleted cells but *PHLPP2* did not change. PTEN mRNA expression was increased in low glucose in HEC-1A cells and both glucose conditions in Ishikawa cells.

We analyzed the correlation between BMI-1 and *PTEN, PHLPP1,* and *PHLPP2* expressions in cells treated with different PTC-209 concentrations (Appendix A). There was a significant inverse correlation between BMI-1 expression and PHLPPs expression but only in hyperglycemia conditions. Thus, it seems that the decrease in AKT phosphorylation in BMI-1-depleted cells may be caused by the reduced expression of PHLPPs but only in hyperglycemia conditions.

### 3.3. Effect of Insulin on BMI-1 Dependent Regulation of AKT Pathway

Our results showed that insulin stimulation impact differently on BMI-1 expression in HEC-1A and Ishikawa cells. In HEC-1A, both in hypoglycemia and hyperglycemia conditions, insulin stimulation caused at first an increase in BMI-1 protein level and then after prolonged stimulation the level of BMI-1 decreases. The level of protein after 8 h insulin stimulation was even lower than in unstimulated cells (Figure 3A–C). In low glucose, the decrease in the BMI-1 level was accompanied by an increase in PTEN and a decrease in AKT phosphorylation. In high glucose, the increase in PTEN was not observed in cells stimulated with insulin for a longer time, but, still, there was a decrease in the pAKT level in cells stimulated with insulin for 8 h.

In Ishikawa cells, the amount of BMI-1 protein showed a slight increase after prolonged stimulation with insulin and did not decrease after 8 h stimulation as in HEC-1A cells. (Figure 3D–F). The pAKT level was increased after 2–4 h stimulation and then slightly decrease, but it was not lower after 8 h stimulation than in control cells.

The analysis of PHLPP phosphatases mRNA expression showed that prolonged insulin stimulation of HEC-1A cells caused both in hypoglycemia and hyperglycemia conditions to a decrease in *PHLPP1* and increase in *PHLPP2* after 4 h stimulation and then the level was similar to in control cells (Figure 4A). In cells treated with PTC-209, the effect of insulin on the *PHLPP2* increase was significantly enhanced (Figure 4B). A similar effect was seen in Ishikawa cells, except that additionally, insulin stimulation also caused an increase in *PHLPP1* but only in hypoglycemia conditions (Figure 4C,D). These results suggest that prolonged insulin stimulation may cause a decrease in AKT phosphorylation by BMI-1 expression reduction, which impacts *PTEN* and *PHLPP2* in HEC-1A cells. In Ishikawa cells that lack PTEN protein, the role of both PHLPP1 may be more pronounced in hypoglycemia conditions.

### 3.4. PTC-209 Inhibits Endometrial Cancer Cells Proliferation, Migration, and Invasion

The effects of PTC-209 in hypoglycemia and hyperglycemia on the proliferation of endometrial cancer cells are shown in (Figure 5A,B). The viability of cells was higher in high glucose compared to low glucose conditions. PTC-209 significantly inhibited cell viability in a dose-dependent manner in cells grown in both glucose conditions. However, the Ishikawa cells seem to be less sensitive to PTC-209 in hypoglycemia conditions compared to HEC-1A cells.

The effect of the PTC-209 inhibitor on the migration capacity of endometrial cancer cells using a wound-healing assay and Transwell assay. After the injury of the tipping scratch, the HEC-1A and Ishikawa cells were treated with PTC-209. The cell migration ability was estimated based on a measure of distance between the wound edges after 0, 24, and 48 h. We observed a significantly reduced migratory potential of cells treated with PTC-209 compared to the control (Figure 6A). Similar results were also detected in the Transwell assay for both cell lines (Figure 6B). The invasion ability was estimated using Transwell assay with Matrigel. The results showed decreased invasion ability of HEC-1A and Ishikawa cells after PTC-209 treatment (Figure 6C). To evaluate the effect of BMI-1 inhibition on epithelial to mesenchymal transition, we examined the expression of several genes involved in these processes such as *SNAIL*, *SLUG*, *ZEB1*, *TWIST*, and *CDH1*. The results demonstrated that PTC-209 treatment was associated with a decreased transcript level of *SLUG* in HEC-1A cells (Figure 6D) and decreased transcript level of *SLUG* and *SNAIL* in Ishikawa cells (Figure 6E). In both cell types, the expression of *CDH1* was significantly increased in both cell types (Figure 6D,E). These results suggest that BMI-1 inhibition by PTC-209 might impact genes involved in epithelial to mesenchymal transition in endometrial cancer cells and, by that, affect the cell migration and invasion ability of endometrial cells.

## 4. Discussion

Alterations in the AKT pathway are common in endometrial cancers but the role of BMI-1 in AKT activity regulation is not yet fully recognized. The results of our previous study showed that, in endometrial cancers, there was a correlation between BMI-1 expression and the level of AKT phosphorylation [14]. That was not surprising because, in many cancer cell types, BMI-1 downregulation causes inhibition of the AKT pathway. Song et al. (2009) showed that, in nasopharyngeal cancer cells, BMI-1 transcriptionally downregulated the expression of the tumor suppressor PTEN through direct association with the PTEN locus [12]. Similarly, silencing of BMI-1 in melanoma cancer cells A375 caused an increased expression of PTEN and reduction in AKT phosphorylation [17]. However, in contrast to these findings in lung cancer cells, A549 BMI-1 did not affect PTEN expression or AKT phosphorylation [19]. Thus, the impact of BMI-1 on the AKT pathway may be cell-context dependent. Moreover, the question is how BMI-1 affects AKT phosphorylation in PTEN negative cells. Previously we analyzed the influence of BMI1 silencing on the expression of several phosphatases involved in the direct and indirect regulation of AKT phosphorylation in HEC-1A cells and showed that BMI-1 downregulation affects *PHLPP1* and *PHLPP2* [14]. There were also significant inverse correlations between BMI-1 and PHLPPs, especially *PHLPP1*, in normal and cancer endometrial tissue samples [14].

In this study, we analyzed the impact of PTC-209, which is a novel inhibitor of BMI-1 translation on AKT phosphorylation and PHLPPs expression in cells differing in the PTEN status grown in different glucose concentrations and insulin stimulation conditions. Studies by Elango et al. (2019) established targeting the insulin signaling pathway as a potential mechanism by which PTC-209 inhibited tumorigenicity of the MDA-MB-231 model [22]. In our study, we wanted to verify our hypothesis that depending on glucose availability and molecular context, BMI-1 can differently affect the activation of the AKT pathway by regulation of PTEN or PHLPP isoform expression in endometrial cancer cells.

We used Ishikawa cells since, despite the presence of PTEN transcript, they lack functional PTEN proteins [21]. Our studies showed that PTC-209 is equally effective in the reduction in the BMI-1 protein level as RNAi interference and causes the same changes in phosphatases levels. The reduction in the BMI-1 level caused a significant increase in expression of *PHLPP1/2* and a decrease in AKT phosphorylation in both cell types. This suggests that BMI-1 may affect the AKT activity not only by PTEN regulation but also by the regulation of PHLPP genes.

Molina et al. (2011) found out that, in glioma cells, the inhibition of both PTEN and PHLPP caused a significant increase in AKT activity while inhibition of only one of two phosphatases did not cause such a profound effect [23]. However, in PTEN depleted cells, the role of PHLPP1 was more pronounced and loss of this phosphatase definitely caused an increase in AKT phosphorylation. Thus, it seems that in PTEN-depleted endometrial cancer cells, PHLPP phosphatases may play an especially important role in AKT regulation. BMI-1 directly regulates the PHLPP expression, which was confirmed by BMI-1 binding to PHLPP genes promoter regions, and, in both cell types, its role seems to be the same.

Epidemiological studies suggest that obesity, hyperglycemia, and hyperinsulinemia are important risk factors for endometrial and breast cancers. Insulin resistance is strictly associated with more advanced cancers and a worse prognosis for breast and endometrial cancer patients [2]. It has been found that insulin resistance is associated with a higher FIGO stage of endometrial cancer and metastases occurrence [24]. Increased expression of PTEN negatively impacts insulin sensitivity and may cause insulin resistance in insulin-dependent cells [25]. A high level of insulin in the blood may stimulate the insulin receptor in different tissues, for example endometrial tissue [26]. Changes in PHLPP expression are also observed in skeletal muscles and fat tissue in diabetes patients [27]. It is suggested that insulin causes a decrease in PHLPPs expression and in effect increases AKT phosphorylation. Studies by Caricilli et al. (2012) confirmed that increased expression of PHLPPs in obese mice caused decreased sensitivity to insulin [28]. Cannon et al. (2014) showed that BMI-1 may be involved in insulin resistance because there was an inverse correlation between BMI-1 level and insulin sensitivity of liver and muscle cells in mice [28].

The mechanisms of hyperglycemia and insulin resistance that impact cancer onset and progression are not completely understood. Duan et al. (2019) showed that BMI-1 expression is increased in high glucose concentrations in pancreas cancer cells PANC-1 and SWI1990 [29]. In endometrial cancer cells, the BMI1 transcript level was higher in high glucose concentration but the Bmi-1 protein level was higher in low glucose compared to high glucose. These results suggest that there may be post-transcriptional regulation of the BMI-1 level by a glucose-dependent mechanism. The glucose concentration also affected PTEN and PHLPP1. The effect of BMI-1 inhibition in HEC-1A cells on PTEN and PHLPPs was different in hypoglycemia and hyperglycemia. It seems that in low glucose the lower phosphorylation of AKT seems to be dependent on PTEN while in high glucose more is on PHLPPs expression (Figure 7). The results showed that insulin stimulation of HEC-1A cells at first causes the increase and then decrease in BMI-1 expression both in hypoglycemia and hyperglycemia conditions. Changes in BMI-1 expression are correlated with changes in AKT phosphorylation but decreased expression of BMI-1 during prolonged insulin stimulation is accompanied by increased expression in PTEN only in hypoglycemia conditions. In Ishikawa cells, insulin stimulation causes *PHLPP1* and *PHLPP2* to increase in hypoglycemia conditions and this effect is enhanced by PTC-209 treatment.

Despite that BMI-1 is recognized as an oncogene involved in carcinogenesis, its role in metastasis and invasion is not fully known. It has been found that the overexpression of BMI-1 promotes invasion and metastasis in pancreatic and hepatocellular cancers [13,18]. However, Xiong et al. [19] showed that in the case of lung cancer cells low expression of BMI-1 is associated with metastasis. Engelsen et al. (2008) showed that the lack of or low BMI-1 expression is correlated with invasion and poor prognosis for endometrial cancer patients [20]. Moreover, Shao et al. (2014) suggested that BMI-1 overexpression is associated with a better prognosis for breast cancer patients [30]. Thus, the role of BMI-1 in invasion and metastasis may be dependent on the cancer type and its molecular context. It has been shown that in melanoma and nasopharyngeal cancer, the inhibition of BMI-1 expression caused the inhibition of epithelial—mesenchymal transition of these cells [12,17]. EMT is a complex process in which cells gain the ability for migration and invasion. During EMT, there is an increase in expression of several transcription factors such as SNAIL, ZEB1, TWIST, and SLUG as well as proteins involved in migration, for example, vimentin or N-cadherin. As a result, EMT expression of E-cadherin is decreased. In endometrial cancers, SNAIL and SLUG play a significant role in EMT because they inhibit the expression of the E-cadherin gene (CDH1) [31]. It has been shown that in endometrial cancers the expression of this SNAIL and SLUG is significantly higher compared to normal endometrial tissue. An increase in the expression of these factors was correlated with the stage of the tumor and lymph node metastasis [31]. Our results showed that in both endometrial cancer cells BMI-1 may be involved in EMT because inhibition of BMI-1 by PTC-209 caused a decrease in SNAIL and SLUG expressions and an increase in CDH1. Moreover, a decrease in BMI-1 expression caused the inhibition of migration and invasion potential of endometrial cancer cells.

Thus, our results suggest that BMI-1 may impact the regulation of AKT activity both via PTEN or PHLPPs expression; this is associated with the proliferation and metastasis ability of endometrial cancer cells.

The explanation of BMI-1’s role in insulin signaling and PHLPP regulation may contribute to a better understanding of mechanisms underlying hyperglycemia and cancer progression. The negative regulation of AKT by PHLPP plays an important role in both suppressing and promoting a disease phenotype (cancer vs. diabetes). This indicates the potential importance of tissue-specific PHLPP function, especially in the context of drug development. This needs to be addressed by the identification of different factors involved in cell/tissue-specific PHLPP signaling. Understanding the relationship between hyperglycemia, BMI-1, and PHLPP functions may contribute in the future to the development of therapeutic approaches that will be better adapted to the molecular context and the specificity of endometrial cancers and be more effective in the case of metabolic disease co-existence. However, future studies are needed to identify all players in the glucose- and BMI-1-dependent regulation of cancer onset and progression. It has been shown that, in human pancreatic ductal cells, BMI-1 combines with the KRAS oncogene to induce malignant transformation [32]. The involvement of KRAS in glucose and the energy metabolism of cancer cells is well established and there is growing evidence linking mutations in KRAS and aberrant PI3K/AKT/mTOR pathway activity [33]. KRAS is not so frequently mutated in endometrial cancer as in pancreatic cancer but, still, KRAS mutations are present in 10–30% of type I estrogen-related endometrial cancer [34]. An increase in KRAS expression has been considered a potential prognostic marker since it has been associated with the transition from pre-malignant to malignant cell status, as well as progression from early to more advanced invasive cancer [35]. Thus, in future studies, the broader molecular context should be taken into account.

## 5. Conclusions

In conclusion, our results showed that BMI-1 expression is glucose and insulin-dependent, especially in PTEN-positive cells, and impacts AKT phosphorylation. In cells that lack PTEN protein, the role of PHLPPs seems to be even more pronounced. We think that further studies dedicated to the analysis of the association between BMI-1 and PHLPPs in endometrial or breast cancer patients with diabetes co-existence are needed to shed more light on the clinical significance of this relationship.

## Figures and Tables

**Figure 1 cancers-14-05947-f001:**
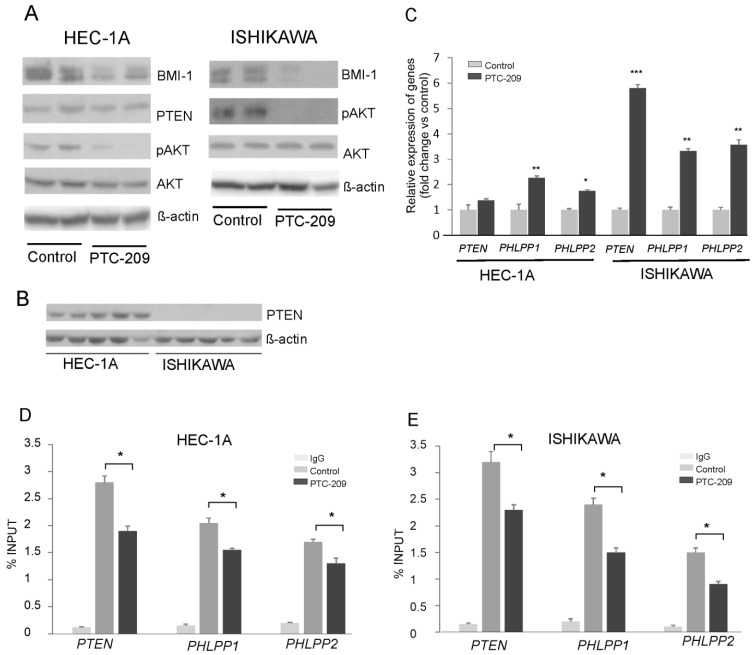
Impact of PTC-209 on BMI-1, AKT, and phosphatases expression. HEC-1A and Ishikawa cells were treated with 5 µM PTC-209 for 48 h and BMI-1 and PTEN protein levels, as well as AKT phosphorylation levels (Ser473), were analyzed by Western blot (**A**). The results of two independent repetitions for control and PTC-209 treated cells are shown in blots. Lane 1, 2: control unstimulated cells; lane 3, 4: cells treated with 5 µM PTC-209. PTEN protein is present only in HEC-1A cells (**B**). Lanes 1–5: HEC-1A cells, lanes 6–10: Ishikawa cells. Changes in *PTEN*, *PHLPP1*, and *PHLPP2* mRNA levels in cells treated with PTC-209 were analyzed using the real-Time PCR method (**C**). ChIP assay was performed to examine the BMI-1 binding to *PTEN*, *PHLPP1*, and *PHLPP2* promoters in HEC-1A (**D**) and Ishikawa (**E**) cells. All the experiments were performed in triplicate and the obtained results are shown as means ± SD; * *p* values of <0.05; ** *p* values of <0.01, and *** *p* values < 0.001. The whole wester blots are showed in Appendix A.

**Figure 2 cancers-14-05947-f002:**
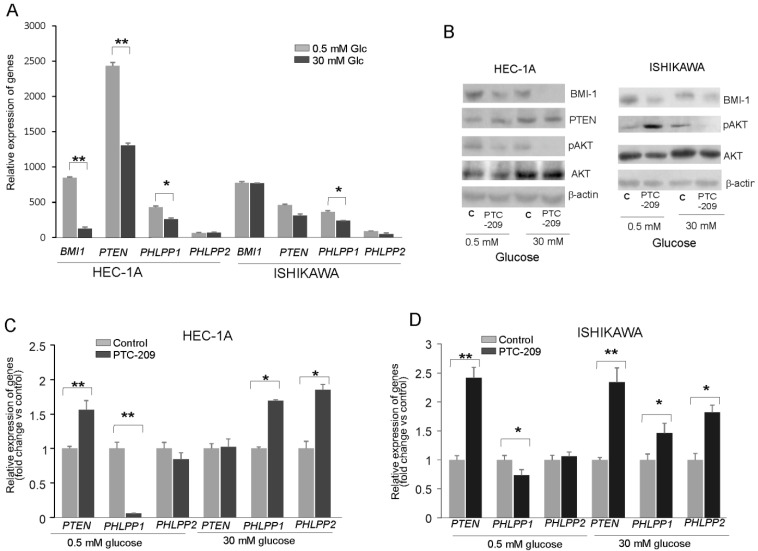
Glucose-dependent effect of BMI-1 inhibition in endometrial cancer cells. (**A**) The mRNA expression level of *BMI-1*, *PTEN*, *PHLPP1*, and *PHLPP2* in HEC-1A and Ishikawa cells grown in hypo- and hyperglycemia conditions. The effect of PTC-209 on BMI-1and PTEN protein–protein levels and AKT phosphorylation level (Ser473) (**B**) as well as PTEN and PHLPP1/2 in HEC-1A (**C**) and Ishikawa (**D**) cells growing in low or high glucose concentrations. The figure shows the means ± SD for three experiments performed in triplicate. The asterisks indicate values of expression that were significantly different in cells treated with PTC-209 compared to control cells; * *p* values of <0.05; and ** *p* values of <0.01. The whole wester blots are showed in Appendix A.

**Figure 3 cancers-14-05947-f003:**
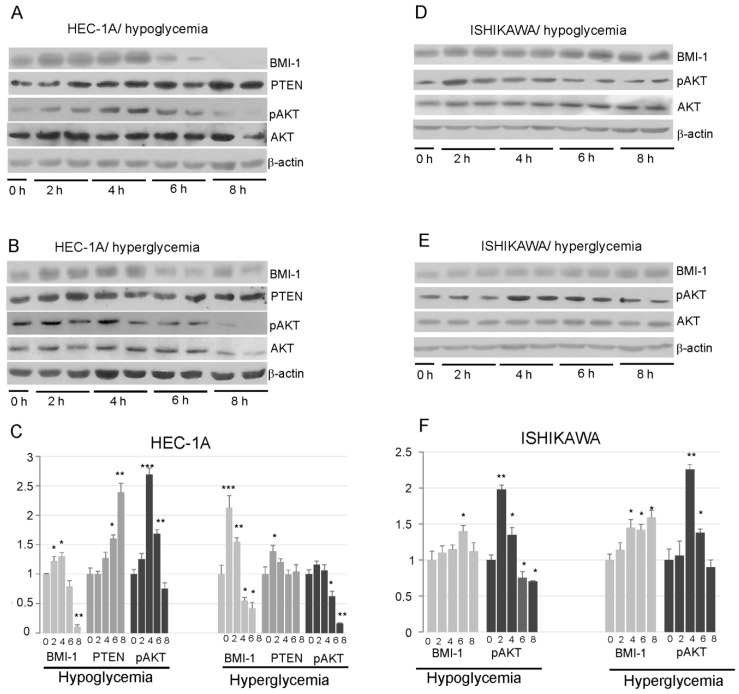
Effect of insulin stimulation on BMI-1, PTEN, and AKT phosphorylation levels (Ser473) in HEC-1A (**A**,**B**) and Ishikawa (**D**,**E**) in low and high glucose concentration. The results from two independent repetitions of insulin treatment for 2, 4, 6, and 8 h are shown in each blot. The intensity of bands corresponding to proteins after Western blot was analyzed by densitometry. The results are shown as the fold change of proteins level after 2, 4, 6, and 8 h of insulin stimulation vs. control (unstimulated cells) in HEC-1A (**C**) and Ishikawa (**F**) cells. The figures show the means ± SD for three experiments performed in triplicate. The asterisks indicate values of expression that were significantly different in cells treated with PTC-209 compared to control cells; * *p* values of <0.05; ** *p* values of <0.01, and *** *p* values < 0.001. The whole wester blots are showed in Appendix A.

**Figure 4 cancers-14-05947-f004:**
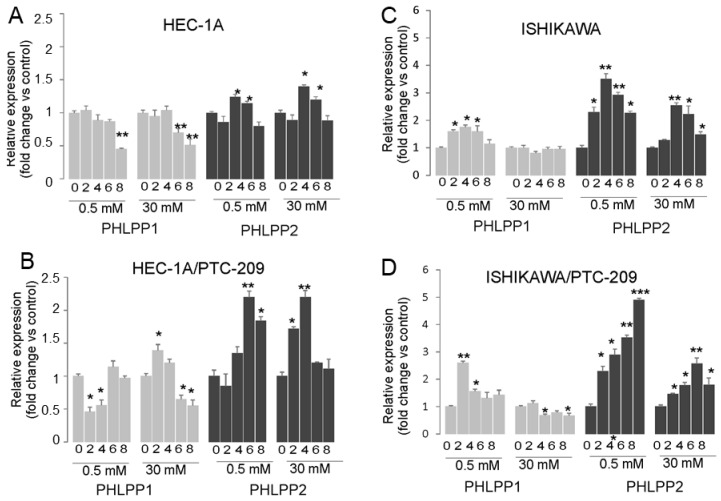
Effect of insulin stimulation on expression of *PTEN*, *PHLPP1,* and *PHLPP2*. HEC-1A (**A**,**B**) and Ishikawa (**C**,**D**) cells were treated with insulin for 0, 2, 4, 6, and 8 h in hypo- or hyperglycemia conditions in the absence (**A**,**C**) or presence (**B**,**D**) of PTC-209 pretreatment. The figure shows the means ±SD for two experiments performed in triplicate. The asterisks indicate values of expression that were significantly different in cells treated with PTC-209 compared to control cells; * *p* values of <0.05; ** *p* values of <0.01, and *** *p* values < 0.001.

**Figure 5 cancers-14-05947-f005:**
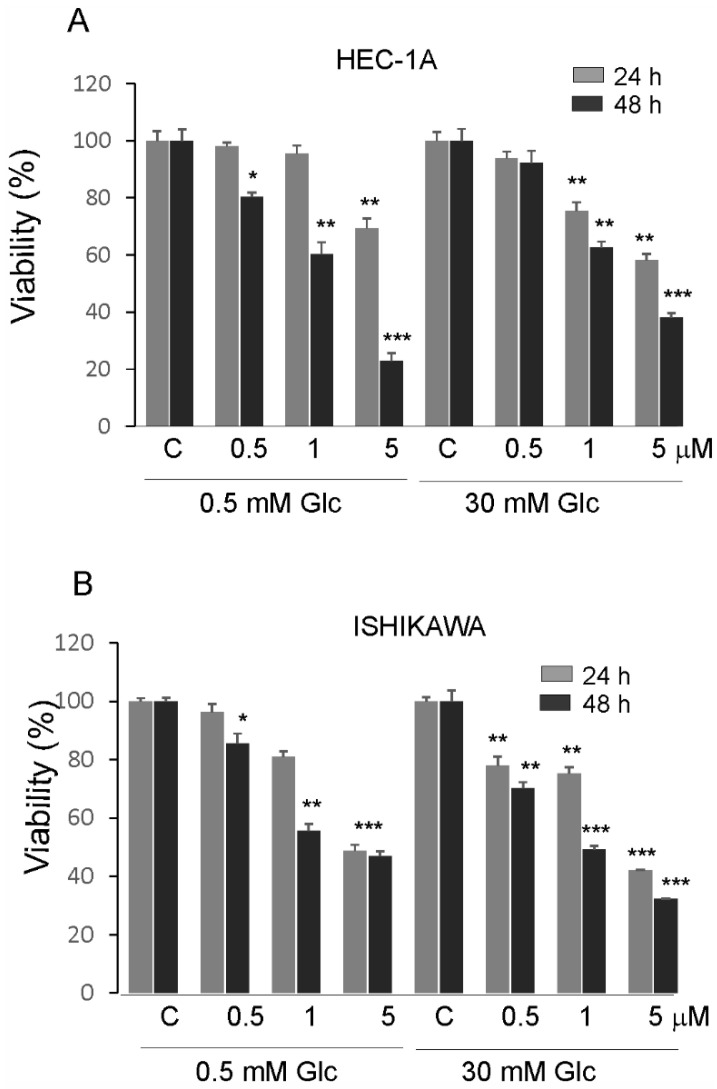
Impact of PTC-209 on the viability of HEC-1A (**A**) and Ishikawa cells (**B**) after 24 and 48 h treatment. The figure shows the means ±SD for three experiments performed in triplicate. The asterisks indicate values that were significantly different in cells treated with PTC-209 compared to control cells; * *p* values of <0.05; ** *p* values of <0.01, and *** *p* values < 0.001.

**Figure 6 cancers-14-05947-f006:**
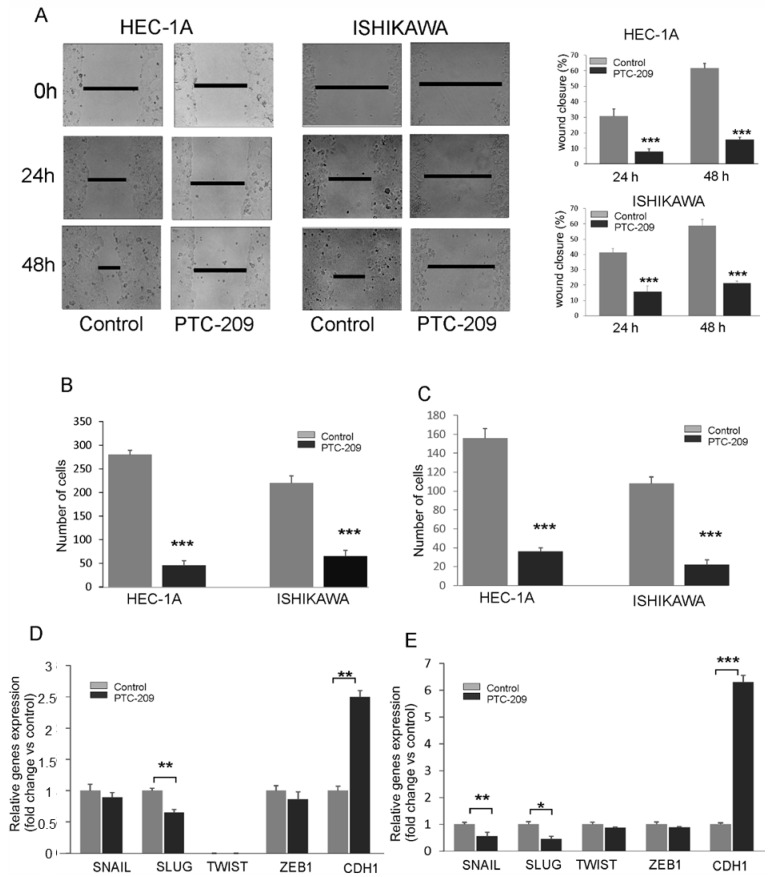
PTC-209 inhibits the migration and invasion potential of cells. Wound healing assay (**A**) and Transwell assay (**B**) were used to estimate the PTC-209 impact on migration potential. To assess invasive chambers were coated with Matrigel^®^ Matrix Basement Membrane (**C**). The impact of PTC-209 on the expression of *SNAIL, SLUG, TWIST1, ZEB1,* and *CDH1* in HEC-1A (**D**) and Ishikawa (**E**) cells was analyzed using time real-time PCR. The figure shows the means ± SD for three experiments performed in triplicate. The asterisks indicate values that were significantly different in cells treated with PTC-209 compared to control cells; * *p* values of <0.05; ** *p* values of <0.01, and *** *p* values < 0.001.

**Figure 7 cancers-14-05947-f007:**
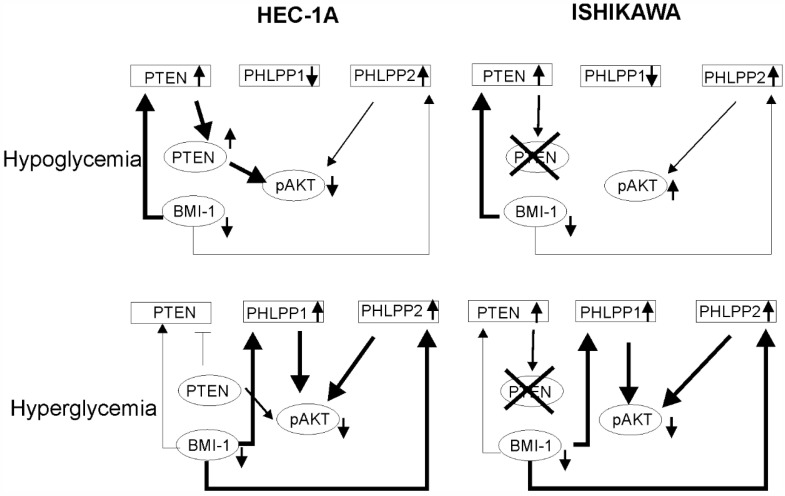
The model of BMI-1’s effect on AKT activation in low and high glucose conditions. In hypoglycemia condition in HEC-1A cells depletion of BMI-1 causes the reduction in pAKT level mostly by PTEN expression increase. In Ishikawa cells, which lack PTEN protein, there is no decrease in pAKT. In hyperglycemia conditions in both cell types, the PHLPPs play a significant role in AKT regulation.

## Data Availability

Data is contained within the article or Appendix A.

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
