# Peer review of "Glucose and Cell Context-Dependent Impact of BMI-1 Inhibitor PTC-209 on AKT Pathway in Endometrial Cancer Cells"

_cancers, 2022, doi:10.3390/cancers14235947_

Round 1
Reviewer 1 Report
In this study Zaczek suggest that BMI inhibitor PTC-209 regulates AKT activity via PHLLP1/2 in a glucose concentration manner. The article is well written and clear. Here are however some concerns that should be addressed:
1) The authors show that BMI-1 expression is almost fully inhibited by PTC-209 in Ishikawa cells (Figure 1A). However CHIP assay detected BMI-1 on PTEN, PHLLP1/2 promoter following PTC209 treatment (Figure 1E). How do the author explain this?
2) PTC-209 concentrations should be given in figure legends. In material and methods it is mentioned that PTC-209 was used at 0.5 1 5 uM, why were different concentrations used?
3) How selective is PTC-209?
4) The activity of AKT is only based on 473 Ser phosphorylation. Previous reports have shown that this residue might in fact rather control the selectivity of AKT for its downstream effectors. Hence some western blots of the phosphorylation of AKT targets should be tested (FOXO, PRAS40, TSC…) to demonstrate that AKT activity was indeed reduced.
5) Migration and survival assay should also be performed with BIM1 siRNA to confirm that PTC-209 effects involve BMI-1
6) PHLLP1/2 levels were only been tested by real time PCR. Some of these results should be confirmed by western blot.
7) Does PHLLP1/2 inhibition prevent the effects of PTC-209 on AKT?
Author Response
The authors show that BMI-1 expression is almost fully inhibited by PTC-209 in Ishikawa cells (Figure 1A). However CHIP assay detected BMI-1 on PTEN, PHLLP1/2 promoter following PTC209 treatment (Figure 1E). How do the authors explain this?
The PTC-209 used in 5 µM concentration very effectively reduced the amount of PTC-209. However, small amounts of protein were still detected in Western blot images (FIG1A). The blots represent the total amount of BMI-1 in cell lysates. We think that a small amount of protein that remains can still bind to chromatin. However, as is shown in figure 1E the binding of BMI-1 after treatment of cells with PTC-209 was significantly reduced compared to control cells.
PTC-209 concentrations should be given in figure legends. In material and methods it is mentioned that PTC-209 was used at 0.5 1 5 uM, why were different concentrations used?
In most experiments, we used a 5 µM concentration of PTC-209, because it gave us the best results of BMI-1 level reduction. However, in some experiments, we used different concentrations, i.e. 0.5, 1, 5 µM to find out if there is an association between the amount of PTC-209 and the viability of cells ( FIG.5 ) as well as whether is the correlation between BMI-1 (reduced by PTC-209) and PTEN or PHLPPs in cells treated with different concentration of PTC-209 (Supplemental Table 2).
How selective is PTC-209?
PTC-209 was originally identified as a low-molecular-weight compound through high-throughput screening using gene expression modulation by small molecules technology (DOI: 10.1038/nm.3418. ) and was reported to exert its anticancer activity through specific targeting of BMI-1 expression. PTC-209 has been reported to interfere with post-transcriptional regulation of BMI-1 and down-regulate BMI-1 production (doi: 10.1038/nm.3418. ). The utility of PTC-209 has been confirmed in many cancers (for example doi: 10.18632/oncotarget.16317; doi: 10.1111/cas.12833, doi: 10.18632/oncotarget.6378 ). The other small-molecule selective inhibitor of BMI1 expression PTC-028/PTC596 has been developed but it exhibits distinct modes of action. PTC-028 induces phosphorylation of BMI-1 at two N-terminal sites, leading to accelerated degradation of BMI-1 (doi: 10.1158/1535-7163.MCT-17-0574; doi: 10.1186/s12943-017-0617-8.). These two compounds are the most selective for BMI-1.
The activity of AKT is only based on 473 Ser phosphorylation. Previous reports have shown that this residue might in fact rather control the selectivity of AKT for its downstream effectors. Hence some western blots of the phosphorylation of AKT targets should be tested (FOXO, PRAS40, TSC…) to demonstrate that AKT activity was indeed reduced.
Thank you for your valuable comments. We are aware that the impact of BMI-1 on AKT and PHLPPs should be studied in more detail in the future. We would like to continue our research. It will be worthwhile in future studies to determine complex mechanisms of associations between BMI-1, PHLPPs, AKT and AKT targets in cells with different molecular backgrounds to have a broader view on relationship between BMI-1 and AKT. In this study, we just wanted to see the effect of BMI-1 dysregulation on AKT and changes in phosphorylation of Ser473 meet our requirements.
Migration and survival assay should also be performed with BIM1 siRNA to confirm that PTC-209 effects involve BMI-1
In the present study, our goal was to assess the effectiveness of PTC-209 in the inhibition of endometrial cancer cells viability and migration potential. However, we agree that it would be worth assessing the impact of BMI-1 silencing on the migratory and survival potential of endometrial cancer cells.
PHLLP1/2 levels were only been tested by real time PCR. Some of these results should be confirmed by western blot.
Anti- PHLPPs antibodies from Abcam company were used during experiments. However, some abnormalities on the Western blot were observed. The antibodies detected a non‐specific band unrelated to PHLPP1, which unfortunately covered the specific one. We reached out to the company and ask them for clarification. They admitted that antibodies have not passed the verification procedure and had been withdrawn from the offer. We did not want to risk wrong interpretation and decided to show results of real-time PCR, of which we are sure. Our decision could be justified by the fact that BMI-1 affects the transcription level of PHLPPs thus mRNA level is better associated with BMI-1 action.
Does PHLLP1/2 inhibition prevent the effects of PTC-209 on AKT?
Thank you very much for your comment. Indeed, investigating whether PHLLP1/2 inhibition will prevent the effects of PTC-209 on AKT would be an interesting aspect of further research. We think that each PHLPPs activity inhibition or silencing in PTEN positive and negative cells would shed more light on PTC-209 action and relationship between BMI-1 and AKT.
Reviewer 2 Report
An article by Dr. Krześlak and the group elaborates on BMI-1 inhibitors' role in glucose metabolism and AKT pathways. This a very well-designed hypothesis-driven research paper that addresses the translational aspects also. But a few things need to be addressed before it is ready for acceptance. They are as follows:
1. It has been shown that BMI-1 and KRAS act as co-occurrence players and induce malignant transformation (PMID: 26951514). At the same time, it has also been discussed oncogenic KRAS-regulated glucose metabolism and AKT-signaling pathways (PMID: 33870211). So it will be worthwhile to determine whether BMI-1 inhibitors also play any role in KRAS-mediated signaling pathways. It will be a future aspect of this current manuscript that authors should discuss in the discussion part as one of the possible future studies by providing the mentioned relevant references.
2. Authors must label all lanes of the western blot. Every western blot has several lanes, but not sure which lane represents what. please label it accordingly.
3. In Figure 4C, the lower part of the figure label is cut. Please fix it.
4. In all western blot figures, wherever applicable please mention which phosphor site of AKT was checked. Mention in the figure labeling- as pAKT S473 or T 308.
Author Response
It has been shown that BMI-1 and KRAS act as co-occurrence players and induce malignant transformation (PMID: 26951514). At the same time, it has also been discussed oncogenic KRAS-regulated glucose metabolism and AKT-signaling pathways (PMID: 33870211). So it will be worthwhile to determine whether BMI-1 inhibitors also play any role in KRAS-mediated signaling pathways. It will be a future aspect of this current manuscript that authors should discuss in the discussion part as one of the possible future studies by providing the mentioned relevant references.
Thank you very much for your valuable comment. We have added the suggestion of further research directions to the discussion section.
Understanding the relationship between hyperglycemia, BMI-1, and PHLPP functions, may contribute in the future to the development of therapeutic approaches that will be better adapted to the molecular context and the specificity of endometrial cancers and more effective in case of metabolic diseases co-existence. However, future studies are needed to identify all players in glucose- and BMI-1-dependent regulation of cancer onset and progression. It has been shown that in human pancreatic ductal cells, BMI-1 combines with the KRAS oncogene to induce malignant transformation [34]. The involvement of KRAS in glucose and energy metabolism of cancer cells is well established and there is growing evidence linking mutations in KRAS and aberrant PI3K/AKT/mTOR pathway activity [35]. KRAS is not so frequently mutated in endometrial cancer as in pancreatic cancer but still, KRAS mutations are present in 10-30% of type I estrogen-related endometrial cancer [36]. An increase in KRAS expression has been considered a potential prognostic marker since it has been associated with the transition from pre-malignant to malignant cell status, as well as progression from early to more advanced invasive cancer [37]. Thus in future studies, the broader molecular context should be taken into account.
Authors must label all lanes of the western blot. Every western blot has several lanes, but not sure which lane represents what. please label it accordingly.
We have added description of the lanes.
In Figure 4C, the lower part of the figure label is cut. Please fix it.
The figure has been fixed.
In all western blot figures, wherever applicable please mention which phosphor site of AKT was checked. Mention in the figure labeling- as pAKT S473 or T 308.
We have added information about the phosphorylation site in the figures.
Round 2
Reviewer 1 Report
No further suggestions
Reviewer 2 Report
All concerns addressed, ready for acceptance.